# Phytochemical Profile and Antioxidant Capacity of *Viscum album* L. Subsp. *album* and Effects on Its Host Trees

**DOI:** 10.3390/plants11223021

**Published:** 2022-11-09

**Authors:** Eva Kleszken, Cornelia Purcarea, Annamaria Pallag, Floricuta Ranga, Adriana Ramona Memete, Florina Miere (Groza), Simona Ioana Vicas

**Affiliations:** 1Doctoral School of Biomedical Science, University of Oradea, 1 Universitatii Street, 410087 Oradea, Romania; 2Department of Food Engineering, Faculty of Environmental Protection, University of Oradea, 26 Gen. Magheru Street, 410048 Oradea, Romania; 3Faculty of Medicine and Pharmacy, University of Oradea, 10 P-ta 1 December Street, 410073 Oradea, Romania; 4Department of Food Science, University of Agricultural Sciences and Veterinary Medicine Cluj-Napoca, 3-5 Mănăstur Street, 400372 Cluj-Napoca, Romania

**Keywords:** *Viscum album* L. subsp. *album*, host trees, flavonoids, antioxidant capacity, proline, chlorophylls

## Abstract

*Viscum album* L. subsp. *album* is a hemiparasitic plant that is recognized as a medicinal plant due to its beneficial effects, including anti-tumor activity, antioxidant, anti-inflammatory, anti-hepatotoxic, hypoglycemic, and antimicrobial properties as well as for lowering blood pressure. On the other hand, mistletoe is a biotic stressor for both deciduous trees and conifers. Our main aim was to evidence the influence of mistletoe on the content of chlorophylls, proline, total phenols, flavonoids, and antioxidant capacity of leaves from tree host trees (*Malus domestica*, *Prunus domestica,* and *Populus alba)* that grow on the northwest of Romania. In addition, HPLC-DAD-MS-ESI+ was used to analyze the phenolic acid and flavonoid profiles of *V. album* L. subsp. *album* leaves according to their parasitized hosts. A significant decrease in chlorophyll *a* level of approximately 32% was detected in poplars infested with mistletoe, followed by infested apples and plums with pigment reductions of 29.25% and 9.65%, respectively. The content of total phenols and flavonoids in the parasitized trees was higher compared to the non-parasitized ones. In the case of poplar, which presented the highest incidence of mistletoe infestation (70.37%), the content of total phenols in the leaves was two times higher compared to non-infested leaves. Based on HPLC chromatographic analysis, leaves of mistletoe growing on apple (VAM) had the highest content of phenolic acids (7.833 mg/g dw), followed by mistletoe leaves on poplar (VAO) and plum (VAP) (7.033 mg/g dw and, respectively, 5.559 mg/g dw). Among the flavonols, the predominant component was Rhamnazin glucosides in the amount of 1.025 ± 0.08 mg/g dw in VAO, followed by VAP and VAM (0.514 ± 0.04 and 0.478 ± 0.04 mg/g dw, respectively). Although our results show that mistletoe negatively influences the host trees, it is still a valuable plant that must be exploited to bring benefits to human health.

## 1. Introduction

*Viscum album* L. known as European mistletoe, is a hemiparasitic, perennial, evergreen plant of the order Santalaces, *Santalaceae* family, ubiquitous in temperate and tropical humid zones and absent in very cold regions, because, under unfavorable conditions, mistletoe seeds are in a state of rest that lasts on average 5–6 months [1].

The Viscum subspecies found in Europe can be distinguished according to the host trees on which it grows. Thus, *V. album* subsp. *album*, grows on deciduous trees and shrubs (*Carpinus betulus, Populus nigra, Salix alba, Tilia cordata, Acer pseudoplatanus, Betula alba, Quercus robur, Ulmus carpinifolia, Craetugus monogyna, Malus domestica, Pyrus communis,* etc.) [2,3]. *V. album* subsp. *austriacum* is found only on the genera Pinus and Picea, *V. album* subsp. *abietis* grows exclusively on the fir and *V. album* subsp. *creticum* is only found in Crete where it parasitizes *Calabrian pine* [1,4]. *V. album* L. also lives on various fruit trees including, but not limited to, apple, plum, pear, and cherry [5]. Mistletoe is a long-term biotic stressor for infested host trees [6].

This semi-parasitic plant acquires mineral salts and water from the host plant, while organic carbon is partially provided by its own photosynthetic activity, as it contains all photosynthetic pigments in its leaves [7]. Water and minerals are acquired by mistletoe directly from the host xylem through its haustorium, with a passive nutrient absorption mechanism. During conditions of physiological stress of the host trees, mistletoe can also actively take up nutrients and water from the host phloem [8].

Studies of the literature have shown that mistletoe can affect the host tree by reducing growth, damaging the quality and quantity of wood, and increasing susceptibility to attack by other pathogens, such as fungi or insects [1]. This exacerbates the effects of water stress and limited resources on host trees, leading to increased mortality [9].

There is no consensus regarding the consequences of tree parasitism by mistletoes. Some studies highlight the negative effects of parasitism by associating them with a reduced photosynthetic activity [10], reduced chlorophyll content [11], biomass loss [12,13], reduction in the amount of absorbed carbon—thereby reducing by 22–43% the carbohydrate content of the host trees [14], sensitization of the trees to the attack of phytopathogenic agents [15], and reduced seed production [16]. Other studies show that with light infestations, the number of dead host trees was lowered [17].

From a biochemical point of view, the parasitism of some Drupaceae species by mistletoe caused a significant reduction in water content and total ascorbic acid content but was not limited to significantly affecting the total concentrations of amino acids, glucose, fructose, and total chlorophyll [18].

The phytochemical composition of mistletoe (*V. album* L.) as well as its in vitro and in vivo effects depend on the host tree [19,20,21,22]. In a recent study, Jäger et al. (2021) investigated the phytochemical profile of leaves, stem and berries of mistletoe growing on three host trees (apple, oak and elm) and highlighted that arginine, pipecolic acid or lysine, dimethoxycoumarin and sinapyl alcohol could be consider as host specific *V. album* biomarkers [19].

*V. album* L. contains a variety of compounds with biological and pharmacological properties. This specie can be used as a complementary remedy, in the treatment of hypertension, atherosclerosis, osteoarthritis, arthritis, diabetes and even cancer [23,24,25,26,27]. *V. album* L shows hepatoprotective, cardiac, neuro-pharmacological and antioxidant activities [28,29,30]. The therapeutic effect of mistletoe (*V. album* L.) could also be due to the synergistic interactions of the different secondary metabolites present in its leaves [31].

The approach in this study was bidirectional; the first direction consisted in highlighting the way in which the mistletoe affects certain physiological parameters and bioactive compounds content of the host tree and the second followed the effects of the host trees on the phytochemical composition and the antioxidant capacity of the mistletoe.

In this sense, the main objective of this study was to prove the influence of mistletoe (*V. album* L. subsp. *Album*) on chlorophyll pigments, bioactive compounds (total phenols and flavonoids), the proline content, and the antioxidant capacity of the leaves of the host trees: apple (*Malus domestica* Barkh.), plum (*Prunus domestica* L.) and poplar (*Populus alba* L.). These parameters were also investigated in the case of the leaves of trees not parasitized by mistletoe, in order to highlight the existence of a possible stress induced by the presence of mistletoe on the host trees. A second goal of the study was to investigate the impact of host trees on mistletoe phytochemicals and in vitro antioxidant capacity.

## 2. Results and Discussion

### 2.1. The Degree of Mistletoe Infestation in the Area under Study

The growth and development of mistletoe depend on several factors such as (i) the nutrients and water availability of the host, (ii) haustorium-inducing factors, and (iii) parasite–host chemical signals [8]. Neither the climatic conditions nor the heterogeneity and structure of the habitat should be neglected, as they have an essential role in mistletoe growth, distribution and host trees preference [4].

Our study analyzes the relationship between mistletoe incidence and its host trees’ height/circumference in an area of Northwest Romania. Table 1 shows the incidence of mistletoe infestation on apple, plum, and poplar.

Table 1 reports that the height of infested trees is higher than that of uninfested trees, thus indicating that the birds prefer tall trees, and the mistletoe needs to be located where the light are more accessible. Taller trees are better hosts for mistletoe than shorter ones because their root systems are deeper in the soil and have greater access to the groundwater and can provide the mistletoe with a better water supply [32].

The total number of mistletoe-infested trees was 37 out of a total of 70 trees. According to the data obtained, the incidence of mistletoe infestation of host trees was quite high, especially in poplar (70.37%), being the tallest and with the largest trunk circumference. Apple and plum were characterized by a lower percentage of mistletoe-parasitized trees, 45.83%, and 36.84%, respectively. In the study area, the mistletoe-infested apple trees did not show any growth reduction. Instead, among the plums infested with mistletoe, one tree showed a decrease in growth rate, while among the poplars, two trees were observed dead.

The results obtained reflect the relationship between the abundance of mistletoe infestations and the health status of the host trees.

Figure 1 shows three host trees infested with mistletoe (*V. album* L. subsp. *album*), along with mistletoe-free trees in the area.

The growth and development of mistletoe is determined by the quality of the host in terms of total nitrogen content and the chemical links between the host and the parasite [8,33]. In other words, parasitic plants affect and are affected by the host trees physiology [34]. Mistletoe (*V. album* L.) is typically found on dominant trees that have thicker branches and a richer crown due to the higher level of light falling on the upper part of the crown, light that mistletoe also needs. These trees have larger root systems that provide better access to water and mineral salts needed by the parasitic plant as well [35]. The thinning of forests and orchards favors the growth of mistletoe by increasing the level of light that reaches its leaves, thus maintenance of a higher stand density is proper forest management for controlling mistletoe infestation [1].

The mistletoe is considered by Szmidla et al. (2019), as a natural forest element with a role in the development and evolution of ecosystems, which reacts to the increasing stress caused by climate change [1].

Mistletoe possesses a higher transpiration rate and stomatal conductance, suggested to ease access to nutrients from the host xylem. Mistletoe infestation does not directly cause the death of the host species, but induces water stress and strongly reduces the carbon assimilation [8].

### 2.2. Macroscopic and Microscopic Evaluation of Mistletoe Leaves

Table 2 includes the length of mistletoe (*V. album* L. subsp. *album*) leaves on apple (VAM), plum (VAP) and poplar (VAO) trees, harvested in May 2022. The length of the leaves varies from 0.6 to 8.6 cm depending on the host tree. Mistletoe leaves harvested from plum (VAP) have significantly larger sizes (*p* < 0.05) than mistletoe leaves from apple (VAM) and poplar (VAO).

Following the microscopic examination, it can be shown that the upper and lower epidermis is made up of a single layer of cells with thickened walls. The upper epidermis consists of rectangular cells and is covered by a thin cuticle. The lower epidermis has slightly elongated cells. The mesophyll is homogeneous in the VAM sample (Figure 2a), while in VAP and VAO samples the mesophyll is differentiated into lacunar (below the upper epidermis) and palisades (disposed towards the lower epidermis) parenchyma (Figure 2b,c).

Conducting tissue consists of 15–17 vascular bundles in the central area and smaller lateral bundles, all surrounded by cells with thicker walls [36].

At the level of the epidermis, paracytic-type stomata can be observed, with two annex cells in all the studied samples (Figure 3). The lower epidermis has numerous stomata, while the upper epidermis has fewer.

### 2.3. The Photosynthetic Pigments of Mistletoe and Its Host Leaves

The mistletoe contains all pigments necessary for photosynthesis, chlorophyll *a*, as the major pigment, chlorophyll *b*, and carotenoids, as accessory pigments. The site of photosynthesis in plants is predominantly the green leaf, but other parts of mistletoe (e.g., branches, stems, floral parts) can undergo photosynthesis [37]. Green pigment content can be altered by both internal factors and environmental conditions [38].

Table 3 shows the levels of chlorophyll *a*, *b*, and carotenoids (mg/g fw) from mistletoe leaves growing on different host trees (apple, plum, and poplar), from leaves of host trees parasitized by mistletoe and from leaves trees (apple, plum, and poplar) not infested with mistletoe.

The highest content of chlorophyll *a*, *b*, and carotenoids were recorded in leaves of mistletoe harvested from poplar (*Populus alba* L.) compared to mistletoe growing on other host trees. The leaves of trees not infested by mistletoe (M, P, and O) show a higher content of photosynthetic pigments compared to those from infested trees (M + VAM, P + VAP, and O + VAO).

Thus, the content of chlorophyll *a* and *b* was significantly reduced in the case of the leaves from the infested host trees. The content of chlorophyll *b* in the leaves of the host tree infested with mistletoe was lower compared with the leaves of trees not infested.

Photosynthetic pigments are important indicators for determining the physiological characteristics of plants. A reduction in the amount of chlorophyll *a* and *b* in the leaves of mistletoe-infested trees may suggest both biotic and environmental stresses [39,40]. Similar results were obtained by Skrypnik et al., 2021 which show that a decrease in the chlorophyll *a*/*b* ratio is correlated with the high degree of mistletoe infestation of the host tree [41].

Reports in the literature regarding the chlorophyll content of the parasite and the host plant are controversial. Some studies conclude that mistletoe, in Drupaceae, did not affect the host’s chlorophyll *a* content but only chlorophyll *b* [18].

Other studies show that mistletoe has significantly lower chlorophyll levels compared to its hosts [42], or that mistletoe chlorophyll levels are similar to those of the apple, pear, and hawthorn host [43], while plum mistletoe has higher amounts of chlorophyll *a* than chlorophyll *b* [18].

According to our study, a significant decrease in chlorophyll *a* level of about 32%, was obtained in the leaves of poplar trees infested with mistletoe followed by leaves of apple and plum trees infested with 29.25% and 9.65%, respectively.

### 2.4. Proline Content of Leaves

The accumulation of free proline is considered an indicator of plant physiology disturbance due to both biotic (pathogens, parasitic plant) and abiotic (drought, salinity, cold, heavy metals) factors [44]. Proline, a nitrogen compound that increases under stress conditions, acts as an antioxidant by maintaining the redox potential in cells and scavenging reactive oxygen species [45].

The current study investigated the level of proline in leaves of mistletoe parasitizing different host trees (apple, plum, and poplar), in leaves of host trees as well as in leaves of trees not parasitized by mistletoe (apple (M), plum (P) and poplar (O)), and the results are shown in Figure 4.

The proline content of mistletoe leaves from apple and plum fruit trees is higher than that of the leaves of host trees and non-parasitized trees. A very high amount of proline (54.45 µg/g fw) was recorded in the case of the VAP sample. The proline content was significantly higher (*p* < 0.01) in the case of plum leaves infested with mistletoe compared to those on non-infested plum trees. On the other hand, no significant difference in relation to the proline content of apple and poplar leaves with or without infestation was recorded.

Üstüner (2019), reported that plants infested with mistletoe (*V. album* L.) had a higher amount of proline than non-infested plants. Mistletoe, as a biotic stress factor, caused an increase in the amount of proline in the host fruit trees (Braeburn apple, Ankara pear, and Hawthorn) [43]. Based on computational modeling, Signorelli et al., 2014 proposed the mechanism by proline act as a protective agent of plants under stress by scavenging the hydroxyl radical [45]. The existence of a Pro-(P5C)-Pro cycle, in the case of plants subjected to stress, suggests that by consuming NADPH within this cycle, proline contributes to maintaining the redox homeostasis [45].

Compared with uninfested trees, leaves of Braeburn apple, Ankara pear, and Hawthorn trees infested with *V. album* L. showed the highest levels of proline, suggesting that mistletoe caused biotic stress [18,43].

On the other hand, the host tree, through its chemical defense mechanisms that involve molecular signals which induce the biosynthesis of secondary metabolites (phenolic substances, nitrogen compounds) inhibits the formation of haustoria and through the production of lignin, suberin tries to counteract the effect of parasitism through the mistletoe [46,47].

### 2.5. Phenols Content of Mistletoe Leaves and Host Tree Leaves

Most of the published articles presenting European mistletoe are related to the description of their phytochemical compounds and their positive effects on various ailments [48,49,50,51]. A major part of the pharmacological activity of mistletoe is attributed to its protein compounds (lectins and viscotoxins) [52,53]. However, the presence of saponins, tannins, and phenolic compounds such as phenolic acids and flavonoids may also have an important role in the biological effects of mistletoe [21,22,29,54]. The presence of these secondary metabolites in mistletoe is dependent on the host tree on which they grow, although until now it has not been clearly established if the secondary metabolites are biosynthesized by the mistletoe or if they are taken up from the host tree. The quantitative data on the composition of phenolic compounds expressed in mg/g dw from the mistletoe ethanol extracts grown on different host trees (apple plum and poplar) are presented in Table 4, and HPLC chromatograms are in Appendix A. Twenty-one compounds were separated and tentatively assigned based on their retention times, UV absorption spectrum, *m/z* values, and main fragments. Table 4 includes the retention time (R_t_), the UV maxim wavelength (λ_max_), the specific *m/z* [M + H]^+^ values, tentative identification of compounds, their inclusion in the subclasses of phenols, and the mean values of mistletoe leaf compounds.

The phenols compounds identified from mistletoe leaves are divided into one hydroxybenzoic acid (compound **1**), eight hydroxycinnamic acids (compounds **2**–**8** and **11**), and twelve flavonols (compounds **9**, **10**, and **12**–**21**). The major chlorogenic acid isomers found in mistletoe leaves include 3-caffeoylquinic acid (compound **2**) and 5-caffeoylquinic acid (compound **4**), while the 4-caffeoylquinic acid (compound **3**) is present only in VAM and VAP. The presence of chlorogenic acid isomers has been previously reported in *V. album* grown on scots pine [55]. Compounds **5**, **7**, and **8** exhibited sinapic acid in their fragmentation patterns (*m/z* 225) and are characterized as sinapic acid glucoside and cinnamate esters as 3 and 5-Sinapoylquinic acid. Compound **6** presented fragment ions at *m/z* 475, 163, and was identified as Dicaffeoyl tartaric acid. This compound was detected only in leaves of mistletoe grown on apple trees (VAM).

Flavonoids such as quercetin derivatives represented by quercetin rutinoside (compound **9**, rutin, *m/z* 611, 303), Quercetin glucoside (compound **10**, *m/z* 465, 303), and quercetin-O-[hydroxymethylglutaryl] hexoside (compound **12**, *m/z* 609, 303) were identified. Isorhamnetin (compound **21**), monomethoxyflavonol (*m/z* 317), and its derivatives such as isorhamnetin glucoside (compound **13**, *m/z* 479, 317), isorhamnetin-O-[hydroxymethylglutaryl] hexoside (compound **14**, *m/z* 479, 317), isorhamnetin glucuronide (compound **15**), Isorhamnetin-dirhamnosyl-rhamnoside (compound **16**) and isorhamnetin glucosyl-rhamnoside (compound **17**) were identified. Two O-methylated flavonols, rhamnazin glucoside (compound **18**, *m/z* 493, 331) and rhamnazin rutinoside (compound **19**, *m/z* 639, 331) were also identified.

Following the analysis of the phytochemical profile, it can be seen that all phenolic compounds identified are found in the VAM sample, whereas the VAP sample lacked compounds **6** and **9** and the VAO sample compounds **3** and **6**.

Regarding phenolic acids, mistletoe leaves grown on apple trees (VAM) are the richest in phenolic acids (7.833 mg/g dw), followed by VAO (7.033 mg/g dw) and VAP (5.559 mg/g dw). Among, the phenolic acids, hydroxybenzoic acid predominates in all samples, and chlorogenic acid predominates among the hydroxycinnamic acids, being in the proportion of 18.04%, 9.91% and 14.22% in VAM, VAP, and VAO, respectively, of the total phenolic acids. Sinapic acid was the major component of the total hydroxycinnamic acids in VAP (14.11%), compared to the other two samples.

Rhamnazin glucosides were the main component of the class of flavonols in VAO (28.49%), followed by VAP and VAM (14.29% and 13.29%, respectively). Rutin was not identified in VAP, being instead present in an almost double amount in VAO compared to VAM, 8.56%, and 4.75%, respectively. In all mistletoe samples, isorhamnetin derivatives predominated compared to those of quercetin in the following descending order VAO > VAM > VAP.

The richest in flavonols is mistletoe grown on poplar (VAO) (7.136 mg/g dw), followed by VAP and VAM with 3.907 mg/g dw and 3.598 mg/g dw, respectively. VAO has the highest content of total phenols (14.169 mg/g dw) compared to VAM (11.432 mg/g dw) and VAP (9.467 mg/g dw).

The total phenolic and flavonoid contents in *V. album* L. subsp. *album* and its host tree leaves were determined and the results are shown in Figure 5a and b, respectively. The leaves are rich sources of phenolic compounds compared with the other organs of the same plant (seed, fruits) [21,55,56].

The total phenolic content of mistletoe leaves ranges between 498.29 to 630.65 mg GAE/100 g fw, while the content of flavonoids varied between 8346.44 and 9683.44 mg QE/100 g fw.

No significant differences were reported between the phenol compounds in the leaves of trees parasitized compared to those not parasitized by mistletoe in the case of apple and plum samples. Instead, significant differences (*p* < 0.0001) were determined in the case of O + VAO and O samples. The content of polyphenols was approximately two times higher in the case of poplar leaves from the parasitized tree compared with the leaves from uninfested poplar. It should be mentioned that of the three trees studied, the poplar had the highest degree of mistletoe infestation, and the defense mechanism of the host plant also consists of the biosynthesis of bioactive compounds.

Similarly, the highest level of flavonoid was recorded in the case of leaves trees compared with mistletoe leaves. Instead, the flavonoid content of VAM was significantly higher compared with VAP (*p* = 0.0131) and VAO (*p* = 0.0029), while no significant difference was recorded between VAP and VAO.

All leaves of the trees infested with mistletoe had significant differences in terms of flavonoid content compared with the leaves from trees without mistletoe, confirming that flavonoids are compounds involved in the host tree’s defense mechanism.

From the class of phenolic acids, hydroxycinnamic acids were the predominant compounds in all mistletoe samples (VAM, VAP, VAO). The presence of these compounds was also reported in the data from the literature regarding the profile of different mistletoe species. Luczkiewicz et al., 2001 analyzed phenolic acids from mistletoe leaves from six host trees (*Sorbus aucuparia*, *Acer plantanoides*, *Malus domestica*, *Pyrus communis,* and *Populus nigra*, *Quercus robur*) and found that in the extract of mistletoe from apple (*Malus domestica*) the main compound was rosmarinic acid (17.48 mg %), while in mistletoe hosted on poplar (*Populus nigra*) the dominant component was chlorogenic acid (12.34 mg%) [21]. Vicaș et al. (2011), investigated the chemical composition of mistletoe (*V. album* L.) from five host trees: *Acer campestre*, *Fraxinus excelsior*, *Populus nigra*, *Malus domestica* and *Robinia pseudoacacia*. *V. album* hosted by *Fraxinus excelsior* recorded the highest amount of total phenolic acids (108.64 µg/g dw), while mistletoe on *Malus domestica* had the lowest total polyphenols level (39.37 µg/g dw). Additionally, in our study, the highest content of phenolic acids was recorded in the VAM sample, with hydroxycinnamic acids being predominant [22].

The variability of the chemical composition of mistletoe can also be associated with environmental conditions such as temperature, soil, O_2_ and CO_2_ concentration, and harvesting period [2]. The impact of the mistletoe harvesting period on chemical composition and antioxidant capacity was previously evaluated [22,29,57]. The winter period, in the presence of snow and less sunshine, is the best time to harvest the mistletoe in terms of polyphenols and antioxidant capacity [29].

In the current study, the phenolic profile of mistletoe extracts growing on three host trees was evaluated, highlighting differences in terms of the types of phenols and their quantity. Pietrzak, W. et al., 2021, highlighted the fact that the highest content of phenolic compounds and implicitly a high antioxidant activity were closely related to the climatic conditions, the autumn–winter period with less sun being favorable for the higher accumulation of secondary metabolites [29].

Flavonol is another group of compounds found in the mistletoe extract (Table 4). The *V. album* L. subsp*. album* leaves contained a complex mixture of 12 flavonoids predominantly as glycosides, with a low amount of aglycones, quercetin, and isorhamnetin (3′-O-methylquercetin). 7,3′-di-O-methylquercetin glucoside known as rhamnazin glucoside was the predominant flavonol found in all mistletoe leaves tested.

Rhamnazin is found also in other medicinal plants, and from the literature, data has been found to have antioxidant, antitumor, antiviral, and anti-inflammatory effects as well as other biological functions [58].

### 2.6. Antioxidant Capacity of Mistletoe Leaves and Their Host Trees

The antioxidant capacity of mistletoe leaves and tree leaves with or without mistletoe infestation were determined by two methods, DPPH and FRAP, and the results are presented in Figure 6.

The reducing capacity of mistletoe extracts may be a good indicator of their potential antioxidant power [59]. The reducing power of mistletoe depends on the host trees, in the case of our study the mistletoe grown on *Populus alba* (VAO) had the highest antioxidant capacity determined by the FRAP method, followed by VAP and VAM. A significant increase (*p* < 0.0001) in the antioxidant capacity of apple and plum leaves from trees not infested with mistletoe was detected by the FRAP method, and slightly, but not significant increases in the case of the DPPH method. Instead, in the case of the poplar, which presented the highest infestation rate, the antioxidant capacity of the leaves is significantly higher (*p* < 0.0001) compared to non-infested trees.

These results are in agreement with other studies in which it was highlighted that the host tree influences the chemical composition of the mistletoe and thus implicitly its antioxidant capacity [21,22,29].

Free radicals and other reactive species are generated by our body in a normal state by various endogenous systems or following exposure to various physical-chemical conditions or in various pathological conditions. Moreover, chemotherapy treatment acts by producing free radicals and antioxidants play a major role in counteracting them and reducing some side effects. Thus, different products based on the *V. album* are used for the complementary and alternative therapy of cancer [5,60].

## 3. Materials and Methods

### 3.1. Biological Material

The plant material used in our study is represented by mistletoe leaves (*V. album* L. subsp. *album*), but also by the leaves of mistletoe-parasitized or non-parasitized trees. The samples were collected near the town of Sânnicolau de Munte, Bihor County. This locality is located on DJ 767A, at a distance of 44 km from Oradea. The town is located at an altitude of 132 m above sea level, with the following coordinates: 47°18′13″ north and 22°8′12″ east, in the northeast of Bihor County. The natural conditions of the researched area include the typical characteristics of the Western Plain, where the climate is temperate—continental, with oceanic influences generated by the western winds.

The plant material was taken on May 2022, from apple (*Malus domestica*, “Crețesc” cultivar), plum (*Prunus domestica*, “Gras românesc” cultivar), and poplar (*Populus alba* L.) trees, parasitized by mistletoe, and from trees not infested by mistletoe (Figure 1). *Mistletoe* leaves, having the same level of development were collected from three different bushes belonging to each host tree, positioned towards the south. The leaves on the host trees were harvested from the same branch on which the mistletoe is positioned (leaves were taken from three trees for each species). Likewise for the leaves on non-parasitized trees, taking into account the leaves should be oriented to the south. Thus, the samples representing three biological replicates of each species were immediately taken to the laboratory and processed for analysis.

A specimen of the mistletoe (*V. album* L. subsp. *album*) was kept in the Herbarium of the Faculty of Medicine and Pharmacy Oradea, Romania, registered in NYBG Steere Herbarium, PUO 05361 code.

The coding of the samples taken in the study is presented in Table 5.

### 3.2. Determining the Degree of Mistletoe Infestation

The degree of mistletoe infestation of three host trees apple (*M. domestica* Barkh.), plum (*P. domestica* L.), and poplar (*Populus alba* L.) was determined. The height and circumference of the trunk of the infested trees were measured.

To evaluate the intensity of mistletoe infestation, the parameter that reflects the number of mistletoe-parasitized trees relative to the total number of trees in the area was used. The incidence of infestation was determined using the Equation (1) [61]:F (%) = 100 × n/N(1)
F (%) = frequency of occurrence of infestation; n = number of trees infested with mistletoe; N = total number of trees in an area.

### 3.3. Macroscopic and Microscopic Characterization of Mistletoe Leaves (V. album L. Subsp. Album)

The macroscopic examination was carried out in order to establish the morphological characters by visualizing the mistletoe leaves with the naked eye or using a magnifying glass.

The microscopic analysis was conducted using the OPTIKA B-383PL light microscope (SC Nitech SRL, Bucuresti, Romania), equipped with Proview digital camera and software. Cross sections were made at the level of leaves (400×) and the longitudinal section was made at the level of the epidermis (1000×), according to standard methods.

### 3.4. Spectrophotometric Determination of Photosynthetic Pigments Content

For the extraction of photosynthetic pigments (chlorophyll *a* and *b*, carotenoids) 0.5 g of fresh leaves were homogenized at 15,000 rpm for 30′ using Ultra Turrax homogenizers (SilentCrusher M instruments, Heidolph Instruments GmbH & Co, Schwabach, Germany) with 10 mL of 95% cold ethanol following the protocol described by Nayek S. et al., 2014 [62]. The samples were centrifuged at 12,000 rpm for 10 min at 4 °C. A volume of 0.5 mL of supernatant of each sample was mixed with 4.5 mL of 95% cold ethanol and used for the spectrophotometric quantification of pigments (664 nm, 649 nm și 470 nm) using a Shimadzu mini UV–Vis spectrophotometer (Shimadzu 1240 mini UV–Vis, Kyoto, Japan). The photosynthetic pigments concentration (μg/mL) was calculated based on the equations presented by Nayek S. et al. (2014) [62] and re-expressed as mg/g fw.

### 3.5. HPLC-DAD-MS-ESI + Analysis Phenolic Compounds of Mistletoe Leaves

#### 3.5.1. Sample Preparation

To separate and identify bioactive compounds by HPLC, mistletoe leaves from the same host trees were air-dried in the dark, at room temperature, and powdered. AN amount of 0.5 g dry powder sample was mixed with 5 mL ethanol 70%, vortexed for 1 min, followed by 30 min ultrasonic treatment. The extract was stored in the refrigerator at 4 °C for 24h and then was centrifuged at 10,000 rpm for 10 min. The supernatant, containing extracted polyphenols, was filtered through a nylon filter (pore size 0.45 μm) and the injection volume was 20 µL.

#### 3.5.2. Chromatographic Condition

Analysis was carried out using a HP-1200 liquid chromatograph equipped with a quaternary pump, autosampler, DAD detector and MS-6110 single quadrupole API-electrospray detector (Agilent-Techonologies, Santa Clara, CA, USA). The positive ionization mode was applied to detect the phenolic compounds; fragmentor, in the range 50–100 V, were applied. The column was a Kinetex XB-C18 (5 μm; 4.5 × 150 mm i.d.) from Phenomenex, Torrance, CA, USA. The mobile phase was (A) water acidified by formic acid 0.1% and (B) acetonitrile acidified by formic acid 0.1%. The following multistep linear gradient was applied: start with 5% B for 2 min; from 5% to 90% of B in 20 min, hold for 4 min at 90% B, then 6 min to arrive at 5% B. Total time of analysis was 30 min, flow rate 0.5 mL/min and oven temperature 25 ± 0.5 °C.

Mass spectrometric detection of positively charged ions was performed using the Scan mode. The applied experimental conditions were: gas temperature 350 °C, nitrogen flow 7 L/min, nebulizer pressure 35 psi, capillary voltage 3000 V, fragmentor 100 V and *m/z* 120–1500.Chromatograms were recorded at wavelength λ = 280 nm, λ = 350 nm and data acquisition were performed with the Agilent ChemStation software (B.02.01SR2, Santa Clara, CA, USA).

The phenolic compounds were identified by comparing the retention time, UV–Vis absorption and mass spectra with those of the standard compounds and with data from the specialized literature [29,55]. Based on the spectral characteristics of phenolic compounds, the wavelength λ = 280 nm is specific for some phenolic acids, flavanol monomers and polymers, while λ = 320 nm for hydroxycinnamic acids and flavonols [63].

Quantification of phenolic compounds was performed using the calibration curve of pure standards (gallic acid, chlorogenic acid and rutin) at concentrations varying from 1 to 100 µg/mL. The regression coefficients of calibration curves ranged between 0.9937 and 0.9981.

### 3.6. Spectrophotometric Determination of Total Phenols and Flavonoids Content

Total phenols content was determined by the Folin–Ciocalteu method with minor changes [22,64,65]. Briefly, each ethanol extract of leaves (100 µL) was mixed with 1700 µL of distilled water, 200 µL of Folin–Ciocalteu reagent (1:10 dilution (*v/v*), freshly prepared) and 7.5% Na_2_CO_3_ solution. The mixture was incubated for 2 h in the dark at room temperature. The absorbance was measured at 765 nm using Shimadzu mini UV–Vis spectrophotometer and the results were expressed as mg of gallic acid equivalent (GAE) per 100 g fresh weight (fw), using gallic acid as a standard (y = 27.637x + 0.0069, R^2^ = 0.9994).

The total flavonoid content of leaves was determined by the aluminum chloride colorimetric method [66]. Briefly, 1 mL of ethanol leave extract was transferred to a 10.0 mL volumetric flask containing 4 mL distilled water, then 300 µL of 5% NaNO_2_ was added to the flask. After 5 min, 300 µL of 10% AlCl_3_ was added and after 6 min, 2 mL of 1 M NaOH. The flask was filled up with distilled water to obtain exactly 10.0 mL and thoroughly mixed. The absorbance was recorded at 510 nm versus blank. Quercetin was used as standard for the quantification of total flavonoids, and the results were expressed as mg QE (quercetin equivalents)/100g fw (y = 0.8475x + 0.0065, R^2^ = 0.9976).

### 3.7. Proline Determination

The content of proline in the leaves was determined according to Bates et al., 1973; Abrahám et al., 2010 [44,67]. Briefly, 0.5 g frozen leaves homogenized (using a cold pestle mortar, kept on ice) with 10 mL of 3% sulfosalicylic acid. The mixture was centrifuged (NÜVE NF 200 Ankara, Turkey) at 5000 rpm for 5 min at room temperature. The supernatant was mixed with ninhydrin acid and glacial acetic acid in the ratio of 1:1:1 (*v/v/v*) and was then incubated at 96 °C, for 1 h. The reaction was stopped on ice and the chromophore was extracted with 4 mL toluene by vigorous stirring for 15–20 s and the absorbance at 520 nm using toluene as reference was measured with a Shimadzu mini UV–Vis spectrophotometer. The calibration curve was determined with different concentrations of standard proline solution, ranging between 5–100 µg/mL (y = 0.0371x + 0.2685, R^2^ = 0.9805).

### 3.8. Determination of Antioxidant Capacity

#### 3.8.1. DPPH (2,2-Diphenyl-1-picryl-hydrazyl-hydrate) Assay

Radical DPPH scavenging capacity of leaves ethanol extract was determined according to the method of Brand-Williams et al., 1995 [68]. A volume of 100 μL of leaves extract was mixed with 2800 μL freshly prepared 80 μM DPPH methanol solution and incubated for exactly 30 min in dark, at room temperature.

The absorbance was measured at 517 nm and the radical scavenging activity was calculated by equation 2 [69] where A0 was the absorbance of DPPH free radical solution in methanol, A1 the absorbance of the leaves extract.
% Radical Scavenging Activity = [(A0 − A1)/A0] × 100 (2)

The results were expressed as mmol Trolox equivalent (TE)/100 g fw.

#### 3.8.2. FRAP (Ferric-Reducing Antioxidant Power) Assay

The FRAP assay was determined according to Memete et al., 2022 [70]. Briefly, leaves extract (100 µL) was mixed with 2000 µL distilled water and 500 µL FRAP working solution (consisting of 300 mM acetate buffer, pH 3.6; 20 mM FeCl_3_·6H_2_O solution, and 10 mM TPTZ solution in the ratio 10:1:1 (*v/v/v*)) freshly prepared and maintained at room temperature, in the dark for 60 min. The absorbance was measured at 595 nm and the results were expressed as mmol TE/100 g fw.

### 3.9. Statistical Analysis

The samples of each leaf extract were analyzed, and all assays were performed in triplicate. The data of analysis are represented as mean value ± standard deviation (SD). The data were subjected to analysis by one-way ANOVA (Tukey’s multiple comparison test) at *p* < 0.05 significant level.

## 4. Conclusions

The mistletoe plant is an important species that can be viewed from two different perspectives, either as a threat to its host trees (especially those in the forest) or as a valuable medicinal plant. In this study, we highlighted the stress induced by *V. album* L. subsp. *album* on three deciduous trees (apple, plum and poplar). Thus, compared to the leaves of non-infested trees, the leaves of mistletoe-parasitized trees showed significant decreases in the content of photosynthetic pigments. The strongest decrease was obtained in the case of the poplar, the tree that presented the highest frequency of occurrence of infestation. In addition, the increase in the concentration of proline in the leaves of the parasitized trees indicates that the mistletoe acts as a stress factor. Infested trees have higher levels of total polyphenols and flavonoids compared to non-infested ones, indicating a possible role of those compounds in the stress response to mistletoe infestation. On the other hand, host trees play a key role in the phenolic profile and antioxidant capacity of mistletoe. Based on the phenolic profile, it was evident that mistletoe is a valuable source in terms of phenolic acid and flavonoid content. The most abundant phenolic acid was chlorogenic acid, while the most abundant flavonoid was rhamnazin glucosides. Compared to mistletoe harvested from plum and apple, mistletoe leaves from poplar contained the highest levels of flavonols and showed the highest antioxidant capacity. Mistletoe’s phytochemical composition and antioxidant capacity are influenced by the host tree, and a screening of the composition is necessary in order to use it effectively in medicine and pharmacy.

## Figures and Tables

**Figure 1 plants-11-03021-f001:**
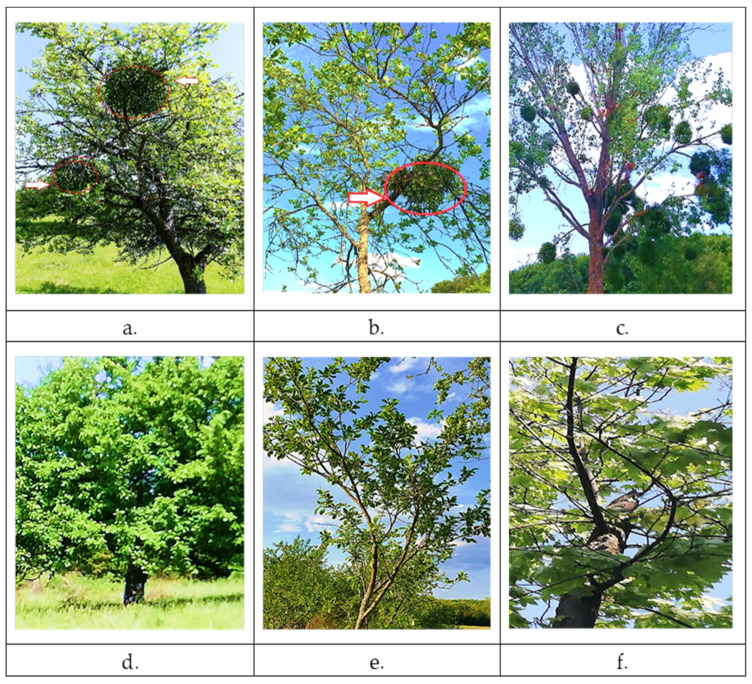
The host trees of mistletoe (*V. album* L. subsp*. album*): (**a**) *Malus domestica* Barkh., (**b**) *Prunus domestica* L., (**c**) *Populus alba* L. Host trees not parasitized by mistletoe: (**d**) *Malus domestica* Barkh., (**e**) *Prunus domestica* L., (**f**) *Populus alba* L., (Location: Sânnicolau de Munte locality, Bihor County, Romania)**.** The arrow and the red circle indicate the presence of mistletoe on the host tree.

**Figure 2 plants-11-03021-f002:**
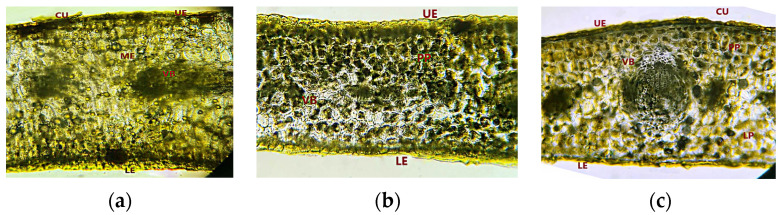
Cross section (400×) of *V. album* L. subsp. *album* leaf growing on (**a**) *Malus domestica* (VAM); (**b**) *Prunus domestica* (VAP); (**c**) *Populus nigra* (VAO); UE—upper epidermis; LE—lower epidermis; CU—cuticle; ME—mesophyll; PP—palisade parenchyma cells; LP—lacunar parenchyma cells; VB—vascular bundle.

**Figure 3 plants-11-03021-f003:**
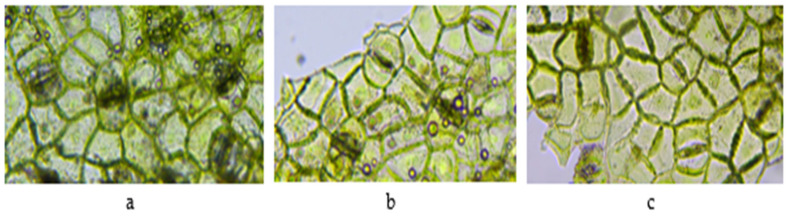
Paracytic stomata in the lower epidermis (1000×). *V. album* L. subsp. *album* leaf growing on: (**a**) *Malus domestica* (VAM); (**b**) *Populus nigra* (VAO); (**c**) *Prunus domestica* (VAP).

**Figure 4 plants-11-03021-f004:**
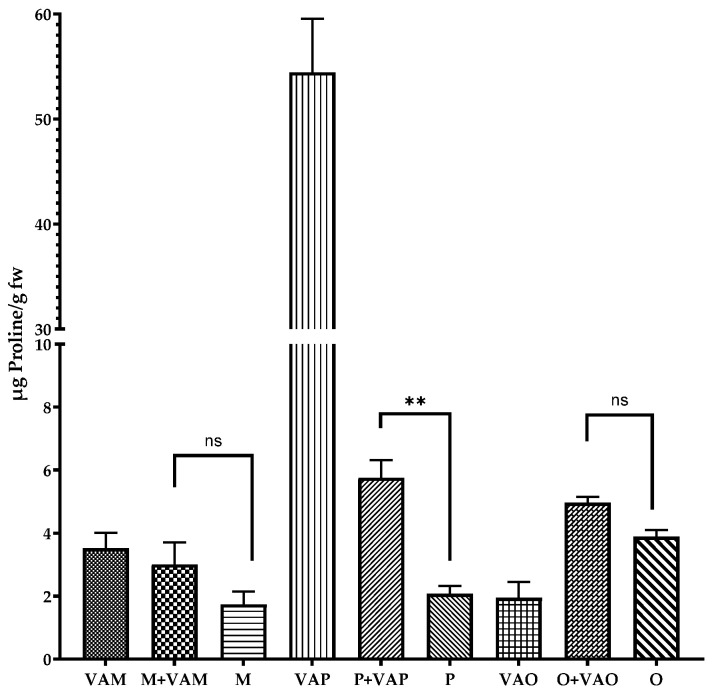
Proline content (µg/g fw) in mistletoe leaves (VAM, VAP, VAO), leaves of host trees parasitized by mistletoe (M + VAM, P + VAP, O + VAO) and leaves of trees not infested with mistletoe (M, P, O). The significant difference was marked only for leaves of trees with or without mistletoe infested; ns—not significantly, ** *p* < 0.01.

**Figure 5 plants-11-03021-f005:**
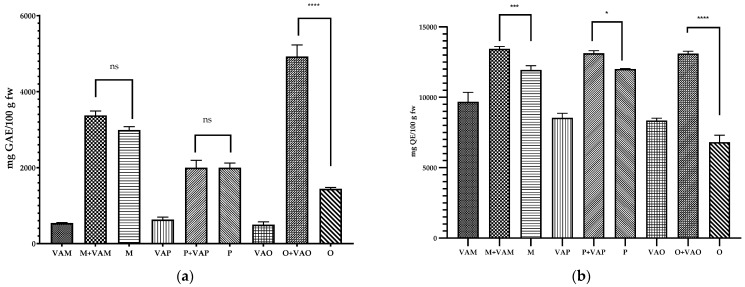
(**a**) Total phenols content. (**b**) Total flavonoid content. Data are expressed as the mean value ± SD (n = 3). The significant difference was marked only for leaves of trees with or without mistletoe infested; ns—not significantly, **** *p* < 0.0001, *** *p* < 0.001, * *p* = 0.0151.

**Figure 6 plants-11-03021-f006:**
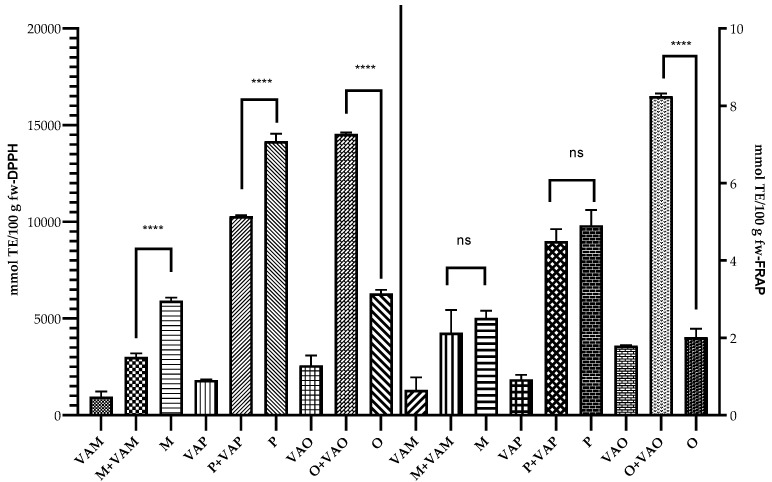
Antioxidant capacity of mistletoe leaves (VAM, VAP, VAO), leaves of host trees parasitized by mistletoe (M + VAM, P + VAP, O + VAO), and leaves of trees not infested with mistletoe (M, P, O). The significant difference was marked only for leaves of trees with or without mistletoe infested; ns—not significantly, **** *p* < 0.0001.

**Table 1 plants-11-03021-t001:** The incidence of mistletoe infestation on apple, plum, and poplar, the height and circumference of infested and uninfested trees.

Trees	F (%)	Infested Trees	Number of Mistletoes on Host Trees	Height of Infested Host Trees (m)	Height of the Uninfested Trees (m)	Circumference of the Trunk of Infested Host Trees (m)	Circumference of the Trunk of Uninfested Host Trees (m)
*Malus domestica* Barkh.	45.83	11	1–5	5.38 ± 1.35 ^c^	4.93 ± 1.80 ^c^	0.85 ± 0.24 ^b^	0.69 ± 0.18 ^b^
*Prunus domestica* L.	36.84	7	1–2	7.61 ± 1.85 ^b^	6.04 ± 1.72 ^b^	0.57 ± 0.15 ^c^	0.54 ± 0.13 ^c^
*Populus alba* L.	70.37	19	3–20	14.01 ± 2.43 ^a^	9.1 ± 0.73 ^a^	1.79 ± 0.68 ^a^	1.51 ± 0.63 ^a^

F (%)-incidence of mistletoe infestation of host trees. Data are expressed as the mean value ± SD. Different superscripts indicate significant differences in the samples (*p* < 0.05) between the height and circumference of the infested and uninfested trees.

**Table 2 plants-11-03021-t002:** Images of mistletoe (*V. album* L. subsp. *album*) leaves grown on apple (VAM), plum (VAP) and poplar (VAO) and their length (cm).

Host Tree	VAM	VAP	VAO
	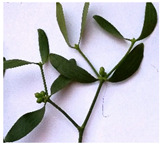	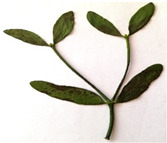	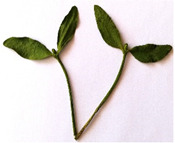
Length leaves * (min–max)	6.13 ± 1.27 ^b^ (0.8–8.6)	7.18 ± 1.15 ^a^ (0.8–10)	4.14 ± 1.18 ^c^ (0.6–4.4)

* Data are expressed as the mean value ± SD (n = 20). Different letters indicate significant difference (*p* < 0.05).

**Table 3 plants-11-03021-t003:** Chlorophyll *a*, *b*, and carotenoid levels (mg/g fw) in leaves *.

Samples	Chlorophyll *a*	Chlorophyll *b*	Total Carotenoids
VAM	1.226 ± 0.005 ^g^	0.493 ± 0.045 ^g^	0.389 ± 0.008 ^g^
M + VAM	1.734 ± 0.016 ^e^	0.670 ± 0.038 ^e^	0.532 ± 0.007 ^e^
M	2.451 ± 0.141 ^c^	1.096 ± 0.086 ^c^	0.674 ± 0.008 ^c^
VAP	1.233 ± 0.012 ^g^	0.497 ± 0.006 ^g^	0.379 ± 0.001 ^g^
P + VAP	2.711 ± 0.006 ^b^	1.042 ± 0.003 ^b^	0.868 ± 0.003 ^a^
P	3.001 ± 0.046 ^a^	1.133 ± 0.109 ^a^	0.848 ± 0.007 ^a^
VAO	1.427 ± 0.023 ^f^	0.607 ± 0.033 ^f^	0.480 ± 0.000 ^f^
O + VAO	1.957 ± 0.059 ^d^	0.889 ± 0.161 ^d^	0.578 ± 0.026 ^d^
O	2.869 ± 0.005 ^a^	1.257 ± 0.011 ^a^	0.706 ± 0.003 ^b^

* Data are expressed as the mean value ± SD (n = 3). Different superscripts indicate significant differences in the samples (*p* < 0.05).

**Table 4 plants-11-03021-t004:** Identification and quantification (mg/g dw) of phenolic compounds from mistletoe leaves samples (*V. album* L. subsp. *album*) from three different host trees *.

Peak No.	R_t_ (min)	UV λ_max_ (nm)	[M + H^+^] (*m/z*)	Compound	Subclass	VAM	VAP	VAO
1	3.33	265	155	Dihydroxybenzoic acid	Hydroxybenzoic acids	2.107 ± 0.19 ^b^	2.695 ± 0.20 ^a^	3.055 ± 0.28 ^a^
2	10.85	323	355, 163	3-Caffeoylquinic acid (Neochlorogenic acid)	Hydroxycinnamic acids	0.775 ± 0.09 ^a^	0.356 ± 0.03 ^b^	0.721 ± 0.06 ^a^
3	11.85	323	355, 163	4-Caffeoylquinic acid (Criptochlorogenic acid)		0.313 ± 0.02 ^a^	0.181 ± 0.01 ^b^	nd
4	12.47	323	355, 163	5-Caffeoylquinic acid (Chlorogenic acid)		1.413 ± 0.11 ^a^	0.551 ± 0.04 ^c^	1.000 ± 0.07 ^b^
5	13.21	330	387, 223	Sinapic acid glucoside		0.490 ± 0.04 ^a^	0.303 ± 0.02 ^b^	0.473 ± 0.04 ^a^
6	13.85	330	475, 163	Dicaffeoyl tartaric acid		0.585 ± 0.05	nd	nd
7	14.28	330	399, 223	3-Sinapoylquinic acid		0.536 ± 0.04 ^a^	0.431 ± 0.03 ^b^	0.016 ± 0.00 ^c^
8	14.84	330	399, 223	5-Sinapoylquinic acid		1.044 ± 0.09 ^a^	0.584 ± 0.05 ^b^	0.663 ± 0.06 ^b^
9	15.75	255, 360	611, 303	Quercetin rutinoside (Rutin)	Flavonol glycoside	0.171 ± 0.01 ^b^	nd	0.308 ± 0.02 ^a^
10	16.18	255, 360	465, 303	Quercetin glucoside		0.323 ± 0.02 ^a^	0.486 ± 0.04 ^b^	0.786 ± 0.06 ^c^
11	16.77	330	225	Sinapic acid	Hydroxycinnamic acid	0.570 ± 0.05 ^b^	0.458 ± 0.03 ^b^	1.105 ± 0.09 ^a^
12	17.01	255, 360	609, 303	Quercetin-O-[hydroxymethylglutaryl] hexoside (Quercetin derivative)	Flavonol	0.392 ± 0.03 ^a^	0.374 ± 0.02 ^a^	0.432 ± 0.03 ^a^
13	17.69	240, 350	479, 317	Isorhamnetin glucoside		0.270 ± 0.01 ^b^	0.232 ± 0.02 ^b^	0.356 ± 0.03 ^a^
14	18.11	240, 350	623, 317	Isorhamnetin-O-[hydroxymethylglutaryl] hexoside (Isorhamnetin derivative)		0.458 ± 0.04 ^b^	0.444 ± 0.04 ^b^	0.830 ± 0.07 ^a^
15	18.77	240, 350	493, 317	Isorhamnetin-glucuronide		0.445 ± 0.03 ^b^	0.390 ± 0.03 ^b^	0.695 ± 0.06 ^a^
16	19.51	240, 350	755, 317	Isorhamnein-(dirhamnosyl)-rhamnoside		0.354 ± 0.03 ^b^	0.368 ± 0.02 ^b^	0.659 ± 0.05 ^a^
17	19.81	240, 350	625, 317	Isorhamnein-glucosyl-rhamnoside		0.249 ± 0.01 ^a^	0.317 ± 0.02 ^b^	0.503 ± 0.04 ^c^
18	20.35	245, 350	493, 331	Rhamnazin glucoside		0.478 ± 0.04 ^b^	0.514 ± 0.04 ^b^	1.025 ± 0.08 ^a^
19	20.71	245, 350	639, 331	Rhamnazin rutinoside		0.279 ± 0.01 ^b^	0.372 ± 0.03 ^b^	0.653 ± 0.06 ^a^
20	21.64	255, 360	303	Quercetin		0.137 ± 0.01 ^b^	0.200 ± 0.01 ^b^	0.677 ± 0.06 ^a^
21	24.03	240, 350	317	Isorhamnetin		0.042 ± 0.003 ^b^	0.210 ± 0.02 ^a^	0.212 ± 0.01 ^a^

nd—not detected. * Data are expressed as the mean value ± SD (n = 3). Different superscripts indicate significant differences in the samples (*p* < 0.05).

**Table 5 plants-11-03021-t005:** The coding of the sample leaves.

Code	Explanation of Coding
VAM	Leaves of *V. album* L. subsp. *album* parasitizing the apple (*M. domestica* Barkh.)
VAP	Leaves of *V. album* L. subsp. *album* parasitizing the plum (*P. domestica* L.)
VAO	Leaves of *V. album* L. subsp. *album* parasitizing the poplar (*P. alba* L.)
M + VAM	Apple leaves from the tree that is infested with mistletoe
P + VAP	Plum leaves from the tree that is infested with mistletoe
O + VAO	Poplar leaves from the tree that is infested with mistletoe
M	Leaves from an apple tree not infested with mistletoe
P	Leaves from a plum tree not infested with mistletoe
O	Leaves from a poplar tree not infested with mistletoe

## Data Availability

Not applicable.

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
