# Peer review of "Phytochemical Profile and Antioxidant Capacity of Viscum album L. Subsp. album and Effects on Its Host Trees"

_plants, 2022, doi:10.3390/plants11223021_

Round 1
Reviewer 1 Report
Several aspects need to be revised in order for the manuscript “Phytochemical profile and antioxidant capacity of Viscum album L. and its effects on their host trees” to be considered for publication, beginning from a typo and an error (subspecies name missing) in the title.
Most of my comments are reported in the attached file; however, I would like to briefly indicate a few major issues
1. It is not clear if the authors have included biological repetitions (which to me are needed in this kind of studies on phytochemicals) other than technical, in their experiments.
2. The microscopy study is lacking accuracy, as far as I can see from the M&M (the authors write about light and electronic microscopy) and Results section (sections artifacts, poor resolution images) and, to me, purpose (afterall, there is not mention of them in the discussion).
3. The discussion is not much more than a literature revision, whereby the authors are supposed to discuss THEIR results in the light of published work.
4. The English language need an accurate revision; too many unclear sentences.

Author Response
We are very grateful for the effort and time you have devoted to this task. We, the authors of the present manuscript wish to thank you for thoughtful commentary you have provided to improve the quality of the paper. We have extensively revised our manuscript according to the recommendations. The whole manuscript was extensively revised and corrections were made throughout.

Reviewer 2 Report
The research paper presents the effects upon the host tree and chemical profile of Viscum album L. (phenolic compounds, flavonoids, antioxidant activity, assimilatory pigments).
Throughout the text "infestation" and "infectation" are used, however infection refers to microorganisms whereas infestation refers to complex organisms. Please verify and modify accordingly.
In lines 118, 122 please use the italic forms for V. album. The same is valid for Figure 2 and Figure 3 text.
Please check the data regarding the flavonoid content (lines 239, 240) because 7.136 g/g seems a bit high and maybe its mg/g. However In line 518 flavonoids were expressed as mg QE/100 g fw.
In line 475, 488, 490 please correct °C because its wrong.
Please rewrite last phrase from line 496-498 because it does not sound good.
Please modify accordingly to the suggestions to improve scientific value.
Author Response
The authors would like to thank you for taking the time to provide us with useful suggestions that improve the quality of the manuscript.

Reviewer 3 Report
The manuscript was well written and it is very original and interesting. Some main remarks: i) the English should be revised by a native English speaker; ii) the abstract should be improved by adding the most important numeric results; iii) the Introduction should be improved by adding more information on the selected tree species (please the authors should better report the reasons for selecting these plants); iv) the plant material should be improved by adding more information on the selected tree species (cultivar?); v) did the authors use validated HPLC methods? vi) the conclusions should be written avoiding redundant information already discussed in the other sections and focusing only on the main results in relation to the aims of the study. A last suggestion: Results and Discussion sections may be integrated into a single section to avoid repetitions.Author Response
The authors would like to thank you for taking the time to provide us with useful suggestions that improve the quality of the manuscript.

Round 2
Reviewer 1 Report
The Authors have done a good job in revising the manucript, which I now consider ready for publication, given they address a few last comments - mainly formal -, that are listed in the attached file.

Author Response
Dear Reviewer 1,
the authors and I in particular would like to THANK YOU very much for taking the time to review our manuscript and for all the comments.
We greatly appreciate the suggestions received which have substantially improved our manuscript.
The answers to the comments are in the attached document.
With respect,
Simona Vicas

Reviewer 2 Report
The overall manuscript quality was improved as suggested by the reviewers.
Corrections were made throughout the manuscript regarding the italic latin names, the correct measure units, and terms in general.
The used methods and results are more clearly represented and the addition of paragraphs gives the manuscript more sense.
The conclusions were rewritten in a more comprehensive manner and avoiding redundant informations already mentioned.
Author Response
Dear Reviewer 2,
thank you very much for your effort to review our manuscript.
Thank you very much for all the suggestions that helped to improve the manuscript.
With respect,
Simona Vicas